# Transcatheter Tricuspid Valve Replacement: Illustrative Case Reports and Review of State-of-Art

**DOI:** 10.3390/jcm12041371

**Published:** 2023-02-09

**Authors:** Manuel Barreiro-Pérez, Rocío González-Ferreiro, Berenice Caneiro-Queija, Marta Tavares-Silva, Luis Puga, Jose A Parada-Barcia, Alvaro Rodriguez-Perez, Jose A Baz-Alonso, Miguel A Pinon-Esteban, Rodrigo Estevez-Loureiro, Andres Iniguez-Romo

**Affiliations:** 1Cardiology Department, Galicia Sur Health Research Institute (IISGS), University Hospital Alvaro Cunqueiro, 36213 Vigo, Spain; 2Cardiac Surgery Department, Galicia Sur Health Research Institute (IISGS), University Hospital Alvaro Cunqueiro, 36213 Vigo, Spain

**Keywords:** structural heart intervention, transcatheter tricuspid valve replacement, tricuspid regurgitation

## Abstract

Tricuspid regurgitation (TR) is one of the most common heart valve diseases, associated a with poor prognosis since significant TR is associated with an increased mortality risk compared to no TR or mild regurgitation. Surgery is the standard treatment for TR, although it is associated with high morbidity, mortality, and prolonged hospitalization, particularly in tricuspid reoperation after left-sided surgery. Thus, several innovative percutaneous transcatheter approaches for repair and replacement of the tricuspid valve have gathered significant momentum and have undergone extensive clinical development in recent years, with favorable clinical outcomes in terms of mortality and rehospitalization during the first year of follow-up. We present three clinical cases of transcatheter tricuspid valve replacement in an orthotopic position with two different innovative systems along with a review of the state-of-the-art of this emergent topic.

## 1. Introduction

Tricuspid regurgitation (TR) is one of the most common heart valve diseases, with a prevalence of 4% in patients ≥ 75 years [1], associated with poor prognosis since the presence of significant TR is coupled with an increased mortality risk compared to no TR or mild regurgitation [2]. Surgery is the standard treatment for TR although in the case of isolated secondary TR, the benefit of surgery over medical treatment in long-term survival may not warrant intervention. In addition, surgery for isolated TR is associated with high morbidity, mortality, and prolonged hospitalization, particularly in case of tricuspid reoperation after left-sided surgery. Thus, several innovative percutaneous transcatheter approaches for repair and replacement of the tricuspid valve (TV) have gathered significant momentum and have undergone extensive clinical development in recent years, with favorable clinical outcomes in terms of mortality and rehospitalization during the first year of follow-up [3]. Also, some imaging advances such as in-silico modeling for preprocedural planning are emerging in the TV field, and are accepted for planning before transcatheter replacement in the aortic or mitral position [4,5]. Between these techniques, transcatheter TV replacement in an orthotopic position is an emergent field. We present three clinical cases with two different innovative systems and a review of the state-of-the-art of this emerging topic.

## 2. Cardiovalve Cases

The Cardiovalve system (Cardiovalve Ltd., Tel Aviv, Israel) is a transcatheter mitral and TV replacement and has received approval for an early feasibility study (NCT04100720) for transcatheter TV replacement (TTVR). Initial experiences have been reported in mitral procedures [6]. The Cardiovalve consists of a steerable transfemoral catheter (32 Fr) and a bioprosthetic valve consisting of three bovine pericardium leaflets sutured using Dacron to a dual (atrial and ventricular) self-expanding fixed nitinol frame for robust radial strength. The valve mends 24 grasping legs for anchoring the device to the native annulus without damage (Figure 1). Three different sizes are provided to expand the annulus from 36 to 55 mm [7].

We describe two cases of transfemoral-transcatheter TV replacement with the Cardiovalve system on two women of 82 and 71 years old, respectively, with permanent atrial fibrillation and torrential TR (Figure 2). The younger patient had undergone a previous rheumatic aortic and mitral surgery whereas the older patient had only atrial functional TR. Both have preserved left ventricular ejection fraction. Nonetheless, they also had moderately reduced right ventricular function (35% measured with cardiovascular magnetic resonance) including 39% for younger patient. A right-heart catheterization was performed in both patients. The 71-year-old patient had an estimated cardiac output of 3.0 L/min with no pulmonary hypertension and elevated right atrial pressure (17 mmHg). The 81-year-old patient had normal cardiac output (4.0 L/min) with pulmonary hypertension (mean pulmonary pressure 28 mmHg) and also higher right atrial pressure (35 mmHg). The two had modestly impaired renal function without cirrhosis. The estimated TRI-SCORE to predict outcome after isolated TV surgery was 14% and 34% for the younger patient and older patient, respectively. Patients were discussed in Heart Team and considered high-risk for conventional surgery. TV repair with the edge-to-edge technique was also debated but taking into account the coaptation gap (>8.5 mm), the TR jet location, and the marked leaflet tethering, neither patient was considered good candidates [8].

A cardiac computed tomography (CT) was carried out in order to evaluate the suitability for TV replacement with Cardiovalve. First, we had to determine tricuspid annulus diameters. A short axis plane at the annulus using multiplanar reconstruction could provide a true assessment of diameters, bearing in mind that maximal dimensions are obtained in late diastole [9] (Figure 3). Right ventricle size needed to be evaluated, and the distance from the annulus to the right ventricular apex had to be over 45 mm. The subvalvular apparatus, especially the anterior papillary muscle, needed to be evaluated; the distance from papillary to the annulus had to exceed 15 mm. Inferior vena cava assessment also had to be performed. The angle between tricuspid annulus and inferior vena cava in long axis view needs to be under 20 degrees to facilitate system coaxially. Finally, a lower limbs venous CT acquisition is mandatory to rule-out any significant stenosis or vascular tortuosity.

Cardiovalve implantation was performed similarly in both patients; we describe the common procedural steps. The procedure is performed under general anesthesia with transesophageal echocardiographic (Figure 4), fluoroscopic (Figure 5), and guidance. A surgical right femoral vein exposition is displayed with a 6 Fr femoral vein sheath exchanged through a 0.035 guidewire for an 18 Fr catheter (Cook Medical, Bloomington, IN, USA). An Agilis Nxt steerable introducer (Abbot, Abbott Park, IL, USA) was used to gain position over the TV and to advance a pigtail catheter to the apex, allowing the advance of an extra-small, high-support Safari guidewire (Boston Scientific, Abbott Park, IL, USA). Guidewire trajectory free of anterior papillary muscle is verified through a 12 Fr reliant balloon (Medtronic, Dublin, Ireland). Next, the 34 Fr Cardiovalve sheath (Cardiovalve Ltd., Tel Aviv, Israel) with the extra-large valve crimped (in both cases) was advanced. The system was guided towards the valve plane by using the control wheels. Once above the valve, the system was raised so that the leaflet grasping legs could be opened in the atrium. Next, the whole system had to be centered and then advanced into the right ventricle in order to grasp the native leaflets. To assure adequate leaflet capture 3-dimensional multiplanar reconstruction was performed with transesophageal echocardiography (TEE). Atrial flange followed by ventricular part were released to complete deployment. The delivery system is centered and retracted towards the right atrium and then the inferior vena cava. The final result was assessed by TEE, ensuring a mean gradient < 1 mmHg and no paravalvular leak or TR. Surgical vein closure was performed.

The older patient suffered a small hematoma at the level of vascular access site and the younger patient had a transient drop in platelet count. Nonetheless, both patients were safely discharged 10 and 7 days after undergoing the procedure, respectively. Nowadays, more than 6 months after the procedure, both patients remain uneventful. No heart failure admissions. Transthoracic echocardiography (TTE) showed no TR, and no paravalvular leak for the younger patient and only a mild paravalvular leak for the older patient.

TV replacement with the Cardiovalve system represents an alternative for patients considered poor candidates for TV repair. The Cardiovalve implant is supplied in three sizes to cover a wide range of native annulus. It also has a very low profile so as not to interfere with the right ventricle structures or the right outflow tract. Additionally, the numerous grasping leaflets provide a solid anchoring accounting with an atrial flange that allows complete sealing, minimizing perivalvular leaks. 

## 3. Lux-Valve Plus Case

The LuX-Valve Plus (Jenscare Biotechnology, Ningbo, China) is an orthotopic transcatheter TV replacement device that is implanted percutaneously through the jugular vein, unlike the first-generation LuX-Valve which is implanted through a minimally invasive right thoracotomy and transatrial approach. The feasibility and efficacy of the first-generation LuX-Valve have been reported by various studies [10,11,12]. The first-in-human study of the LuX-Valve Plus system was published in 2022 and included 10 high-risk patients with more than severe TR [13]. Procedural success was achieved in all patients, without any death or major complication (major bleeding, reoperation or conversion to surgery, stroke, myocardial infarction, or heart failure hospitalization) during the 30-days of follow-up. NYHA class was significantly improved after the procedure and all patients had none/trivial TR at 30-days follow-up. The LuX-Valve was successfully implanted in five patients with previous left-sided valvular surgery and in one patient with a pre-existing permanent pacemaker. This device does not currently have CE mark or FDA approval.

The LuX-Valve Plus system consists of four components (Figure 1): (a) a tri-leaflet prosthetic valve made of bovine pericardium; (b) a self-expandable nitinol valve stent consisting of an atrial disc; (c) an interventricular septum anchoring system, “tongue”; and (d) two expanded polytetrafluoroethylene-covered graspers. The design of the LuX-Valve Plus is radial-force independent, unlike other tricuspid transcatheter valve systems. The prosthesis secures fit by the interventricular anchor hooking to the septum and the two anterior graspers attached to the native valve. The prosthesis is available in three atrial disc sizes: 40, 50, and 55 mm, and in one prosthetic valve size (30 mm). The delivery system is 33 Fr. Baseline echocardiogram, right-heart catheterization, and CT are performed in all patients for an initial screening. The exclusion criteria are systolic pulmonary artery pressure ≥55 mmHg, left ventricular ejection fraction ≤50%, tricuspid annular plane systolic excursion ≤10 mm, right ventricular fractional area change <20%, untreated severe coronary artery disease, Ebstein’s anomaly or arrhythmogenic right ventricular dysplasia, and other valve disease requiring surgery.

We describe a challenging LuX-Valve Plus procedure in a 70-year-old woman diagnosed with familial dilated cardiomyopathy with previous transcatheter mitral valve replacement using Tendyne TMVR system (Abbott Vascular, Santa Clara, CA, USA) and two pacemaker leads (non-functional VVI pacemaker and cardiac resynchronization therapy with defibrillator). The patient is followed-up in the Heart Failure unit and experienced functional class decline (NYHA III; ACC-AHA D) during the last year despite optimal medical treatment, with recurrent hospitalizations due to decompensated heart failure, as well as the need for intermittent ambulatory treatment with levosimendan. Significant recovery of left ventricular ejection fraction occurred after transcatheter mitral valve replacement using the Tendyne TMVR system and a TTE showed progression from severe to torrential tricuspid regurgitation (related to tricuspid annulus dilation and anterior and septal leaflets restriction caused by one of the pacemaker leads placed in-between leaflets), severe right atrial enlargement, and severe right ventricular dilatation (end-diastolic diameter 62 mm) with mild systolic dysfunction (Figure 6). To assess the feasibility of tricuspid valve replacement with the LuX-Valve Plus system, a cardiac computed tomography (Figure 7) was performed to determinate the tricuspid annulus diameters (maximum and minimum diameters), as well as measurement of left atrial and left ventricular dimensions, and to appreciate the full anatomical relationship between the tricuspid valve and the pacemaker leads. Cardiac CT measurements allowed us to select the size of the bioprosthesis and the optimal projection angle for the procedure. The invasive pressure of the pulmonary artery, right atrium, and right ventricular were recorded before LuX-Valve implantation. The patient did not meet any exclusion criteria for device implantation.

The procedure was performed under general anesthesia with intra-procedural imaging guidance using TEE, fluoroscopy, and fluoro-CT fusion imaging (Figure 8 and Figure 9). After percutaneous puncture of the right internal jugular vein under ultrasound control and preimplantation of two Per-close Proglide (Abbott Vascular, Santa Clara, CA, USA), a 7 Fr introducer was placed to advance a stiff Lunderquist guidewire (Cook Medical, Bloomington, IN, USA) into the right pulmonary vein. Subsequently, successive dilations of the venous access were performed with 16, 24, and 33 Fr dilators, and a short hydrophilic introducer sheath (36 Fr) was then placed into the vein and the delivery system was advanced into the right ventricle. The delivery system was placed perpendicular to the annular plane and in a central position, and the outer sheath was gradually withdrawn to release the “tongue”, the valve stent, and the two anterior leaflets graspers. The delivery system was then carefully pulled back, allowing the graspers to capture the anterior leaflet. Then, the atrial disc was released, and the valve started to work. After confirming a satisfactory position and orientation of the valve by echocardiography, the anchoring system was released, and the anchoring needle penetrated the interventricular septum. Finally, the delivery catheter was withdrawn. Hemostasis of the jugular venous access was performed with the two Perclose Proglides combined with a figure-of-eight suture.

LuX-Valve Plus system is a feasible TR treatment and may reflect a safe alternative to conventional surgery in selected high-risk patients to improve symptoms, functional class, and quality of life. Further studies are warranted to provide data on long term outcomes and to evaluate the safety of the device.

## 4. Ttvr Portfolio Review

Nowadays, four dedicated prosthetic valves for orthotopic transcatheter tricuspid replacement have published data on human series or ongoing clinical trials (Table 1).

### 4.1. Navigate

GATE™ (NaviGate Cardiac Structures Inc., Lake Forest, CA, USA) system was the first reported transcatheter tricuspid replacement in 2017 and has, since then, published outcomes [14,15,16]. It is a nitinol self-expanding conical stent with a tri-leaflet equine pericardial valve delivered through a 42 Fr access that can be obtained via transjugular venous access or transatrial puncture after mini-thoracotomy. Anchoring is achieved through grasping of the native leaflets with 12 graspers on the ventricular size and 12 winglets on the atrial side. In a recent cohort of 30 patients, success rate was 87%, with two cases requiring conversion to open-heart surgery and 76% achieving mild or less TR at discharge [15].

### 4.2. Cardiovalve

Cardiovalve (Cardiovalve Ltd., Tel Aviv, Israel) is available for investigational use. Case reports have been published with good clinical outcomes [7,17,18]. It consists of a self-expanding nitinol frame with three bovine pericardial leaflets and has an inflatable cuff solution to minimize paravalvular leaks. The valve is delivered through a 28 Fr transfemoral access and anchored by leaflet grasping. It has four different sizes (S, M, L, XL) up to 55 mm of tricuspid annulus diameter. The valve is designed to treat both TR (transfemoral access) and mitral regurgitation (with transeptal puncture). Clinical data is restricted to case reports but an early feasibility trial with Cardiovalve is currently underway in the U.S. (registered in clinicaltrials.gov, accessed on 10 January 2023, NCT04100720).

### 4.3. Evoque

The Evoque Tricuspid Valve (Edwards Lifesciences, Irvine, CA, USA), like its mitral valve homonym, consists of a self-expanding nitinol frame, bovine pericardial leaflets, and an intra-annular sealing skirt and nine anchors that provide an exclusive anchoring mechanism, using the annulus, leaflets, and chords for secure valve placement. Available in two sizes (44 and 48 mm), it is delivered through transfemoral access with a 28 Fr multiplanar steerable delivery system that facilitates coaxial deployment in most anatomies [19,20]. The first inhuman Evoque transcatheter TV replacement study showed remarkable results at one-year follow-up [20].

Recently, the TRISCEND study that enrolled 176 patients with at least moderate TR showed the procedure was technically feasible with acceptable safety and at 30-day follow-up there was significant reduction in TR with symptomatic improvement [20]. Currently, the TRISCEND II randomized clinical trial is ongoing (registered trial with number NCT04482062).

### 4.4. LuX-Valve 

The LuX-Valve (Jenscare Biotechnology, Ningbo, China) is a self-expanding stented valve, composed by a tri-leaflet bovine pericardium valve, with an external nitinol stent covered with a layer of polyethylene terephthalate. Noteworthy, this valve also possesses an atrial disk, two anterior leaflet graspers, and an interventricular septal anchor that give support to valve placement and attachment, independent of radial forces [11,13]. In the first six implanted valves, technical success was described in all patients without intraprocedural mortality and significant improvement in TR severity. At 12-month follow-up, paravalvular leak did not progress in patients with mild paravalvular leak and substantial improvements in symptoms and NYHA functional class were documented in all but one patient who died due to right ventricular failure [11]. Initially designed to implant through right atrial access via a small right anterior thoracotomy, the second-generation LuX-Valve Plus, presents a new delivery system with an introducer sheath of 33 Fr, prepared for transjugular approach. In the first inhuman study of 10 patients treated with the LuX-Plus valve, procedural success was achieved in all cases, with no intraprocedural mortality or conversion to open surgery. Significant TR reduction and NYHA functional status improvement was observed in all patients at 30-day follow-up [13]. A prospective multicenter study will evaluate the performance and safety of the LuX-Valve Plus system for TV replacement (registered clinicaltrials.gov with number NCT05436028).

### 4.5. Early Experience Phase with Other Devices

The Intrepid Valve (Medtronic Inc, Brooklyn Center, MN, USA) is a dual-stent system with a 27 mm bovine pericardial valve, with an outer frame available in three sizes (43, 46 and 50 mm) initially designed for mitral position [21]. In a compassionate use setting, Intrepid was successfully deployed in three patients with severe TR. The self-expanding nitinol system is delivered through a 35 French sheath, being suitable for transfemoral approach in patients with severe TR [22].

The Novel valve (TriSol Medical, Yokneam, Israel) consists of a self-expanding ni-tinol component frame with a bovine pericardial monoleaflet valve attached by two commissures. With a unique design, in diastole the valve opens with two symmetrical segments opposing on the ventricular face of the annular plane and creating two orifices for antegrade blood flow, while in systole ventricular contraction pushes leaflets back to form a circular line of coaptation with the entire perimeter of the porcine pericardium covered nitinol frame, resembling the closing motion of a mechanical double-disk valve. A first in-human implantation has been described, with successful implantation and complete elimination of TR [23].

The Topaz valve (TRiCares GmbH, Aschheim, Germany) is a new self-expanding bioprosthetic valve of porcine pericardium with a two-stent nitinol frame. The outer stent provides a sealed anchoring into the native tricuspid apparatus while the inner stent houses the tri-leaflet valve, that functions independently of the outer stent, maintaining a circular shape and full valve integrity. The 29 Fr delivery system is designed for transfemoral access. As the anchoring mechanism does not rely on radial force but on a layer of anchors located below the annulus level, no valve oversizing is needed. Currently there is no possibility of recapture during valve deployment and there is only one size available that allows treatment of diastolic annulus diameters <45 mm. The first two inhuman implants reported safety and feasibility of the procedure with excellent immediate hemodynamic results [24].

## 5. Ttvr Clinical Results Review

Up to the time of this review there are three devices with published clinical data in early feasibility studies (NaviGate, Evoque and LuX-Valve systems) and two devices (Evoque and LuX-Valve systems) have reported 1-year follow-up results and have multicentric ongoing trials. Main results are resumed on the Table 2. Other four devices (Cardiovalve, Trisol Valve, Intrepid Valve, and Topaz Valve) have current ongoing feasibility trials.

The NaviGate system (NaviGate Cardiac Structures Inc., Lake Forest, CA, USA) presented in November 2020 a cohort of 30 patients, reporting implantation success in 26 of 30 patients (87%) with conversion to open heart surgery in two (5%), with 76% of patients with mild or less TR at discharge. There were three in-hospital deaths (10%), one uncontrolled bleeding, one multiorgan failure in a patient with baseline cirrhosis and chronic kidney disease, and one after surgical conversion due to malposition of the device. During follow-up (127 ± 82 days), one more patient died due to progressive heart failure, and 62% were in New York Heart Association (NYHA) functional class I or II, with no late device-related adverse events [15]. At the time of this review there are no further trials registered evaluating this device.

The Evoque device (Edwards Lifescience, Irvine, CA, USA) presented in March 2021 a first-in-human experience cohort of 25 patients with tricuspid transfemoral replacement in compassionate use with a technical success of 92%, no conversions to surgery, and no procedural deaths. In this cohort, 36% of patients had pacemaker leads and implantation was successful with either no or mild paravalvular TR at the site of the lead in patients. After a 30-day follow-up, 76% were at NYHA I–II functional class, 90% presented a grade I–II TR, and mortality rate was 0%. The main complications were major bleeding in three (12%) patients, valve reintervention in one (4%), and new conduction abnormalities requiring permanent pacemaker implantation in two (8%) [22]. A one-year follow-up expansion of this trial published in March 2022 reported mortality of 7% (2/27) with 70% of patients in NYHA class I–II and 87% of patients with TR grade I–II, two patients experienced HF hospitalizations, and one patient required a new pacemaker implantation [20].

In March 2022 the TRISCEND trial with the Evoque device presented a cohort of 56 patients in which TR was reduced to mild or less in 98% of them and NYHA class improved to class I–II in 78.8% patients. There were one cardiovascular death in a patient with a failed procedure, two reinterventions after device embolization, one major access site or vascular complication, and 15 severe bleeds, of which none were life-threatening or fatal [25]. A multicentric randomized controlled trial compared with optimal medical treatment (TRISCEND II (NCT04482062)) is ongoing and will conclude its first phase in 2024.

The LuX-Valve system (Jenscare Biotechnology, Ningbo, China), with greater sizes for the annulus (50, 60, 70 mm), presented in May 2021 a first experience with 12 symptomatic patients in NYHA class III or IV with severe or greater TR not suitable for surgery. A transcatheter replacement via minimally invasive thoracotomy and transatrial approach was performed successfully in all cases. There was no intraprocedural mortality nor transformation to open surgery. One patient died at postoperative day 18 due to non-surgery and device reasons. At 30-day follow-up, TR was significantly reduced to grade I or less in 91% patients (10/11) and grade II in 1 patient; NYHA class I–II was achieved in 54.5% patients [12].

In December 2022 a one-year follow-up study with 15 patients with severe or greater TR in NYHA III or IV patients and high surgical risk was presented [26]. The device was implanted via transatrial approach successfully in all patients. At 1-year follow-up TR was significantly reduced to grade I or less in 85.7% and 78.6% were at NYHA class I–II. One patient died on postoperative day 12 of a pulmonary infection. One patient (7.1%) was re-hospitalized during 1-year follow-up because of device thrombosis and two patients had mild paravalvular leakage. A prospective, multicenter trial (TRAVEL trial NCT04436653) is ongoing with the LuX-Valve system and expected to conclude in 2026.

Recently, a minimally invasive second-generation version of the LuX-Valve device with transjugular access, the LuX-Plus system, reported in September 2022 a first-in-human experience with 10 high surgical risk patients, all of them in NYHA class III or IV with implantation success in all of them without in-hospital mortality, major bleeding, or conversion to surgery. In a 30-day follow-up, all patients had reduced TR to grade I or less. One patient (10%) required pacemaker implantation due to third-degree atrioventricular block 2 days after the procedure [13]. Transjugular access shows a promising pathway in order to reduce major access site or vascular complications, LuX-Plus device is currently being evaluated in a single-arm multicenter trial (NCT05436028) that will conclude in December 2023.

Other devices are still in development and lack clinical data at the time of this review. The Cardiovalve system (Cardiovalve Ltd., Tel Aviv, Israel) has presented case reports of successful implantation [7,17]. An early feasibility study with 15 patients is currently ongoing for Cardiovalve transfemoral system (NCT04100720). The Trisol Valve (Trisol Medical, Yokneam, Israel) presented in August 2021 a first human case report with successful implantation of the device [23] and a feasibility trial is registered under the identifier NCT04905017

The Intrepid valve (Medtronic, Minneapolis, MN, USA) has an early feasibility trial ongoing that is expected to be completed in April 2023 (NCT04433065) and the Topaz transfemoral system (TRiCares GmbH, Aschheim, Germany) has an early feasibility trial ongoing (TRICURE NCT05126030).

A limitation of this review is that the information available about these devices is limited, it comes from first-in-human experience or feasibility trials in many cases, and short follow-ups concerning only a few patients have been published. Nowadays, there is not enough experience to position one of the designs over the others, or to indicate its implantation in the clinical setting out of the trials or the compassionate use.

## 6. Conclusions

Transcatheter TV interventions are an appealing treatment option for a common disease in a group of patients often with co-morbidities, high surgical risk, and poor surgical outcomes when not addressed at an early stage. Particularly, transcatheter TV replacement with orthotopic prosthetic valves offers a definitive treatment maintaining the physiological functions of right heart chambers (as opposed to heterotopic prosthesis) and the possibility of treatment of patients with contraindications to other percutaneous interventions such as edge-to-edge repair or annuloplasty devices.

## Figures and Tables

**Figure 1 jcm-12-01371-f001:**
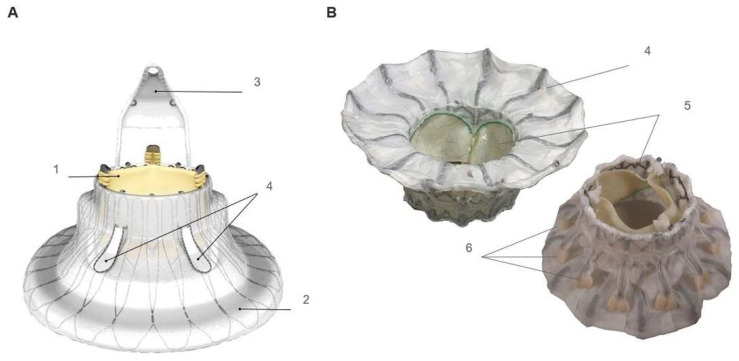
Components and design details of LuX-Valve Plus (**A**) and Cardiovalve (**B**). LuX-Plus Valve: prosthetic valve (1), atrial disc (2), interventricular septal anchor (3), and two graspers (4). Cardiovalve: Atrial flange (4), prosthetic valve (5), and leaflet graspers.

**Figure 2 jcm-12-01371-f002:**
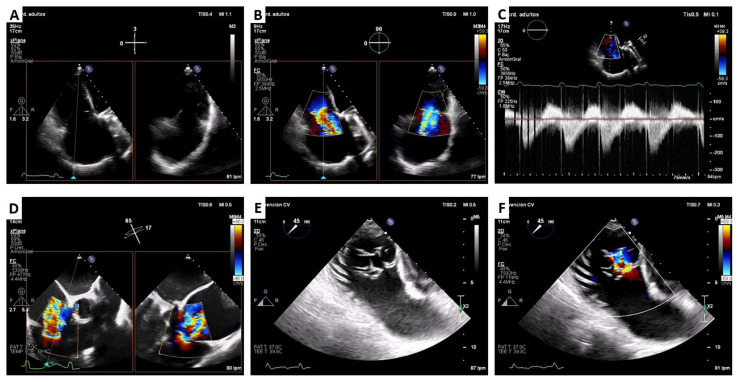
Cardiovalve TTVR case. Pre-procedural transthoracic and transesophageal echocardiography. Massive tricuspid regurgitation: central valve coaptation gap (**A**,**E**), wide vena contracta regurgitation jet (**B**,**D**,**F**), and triangular dense continuous doppler trace (**C**).

**Figure 3 jcm-12-01371-f003:**
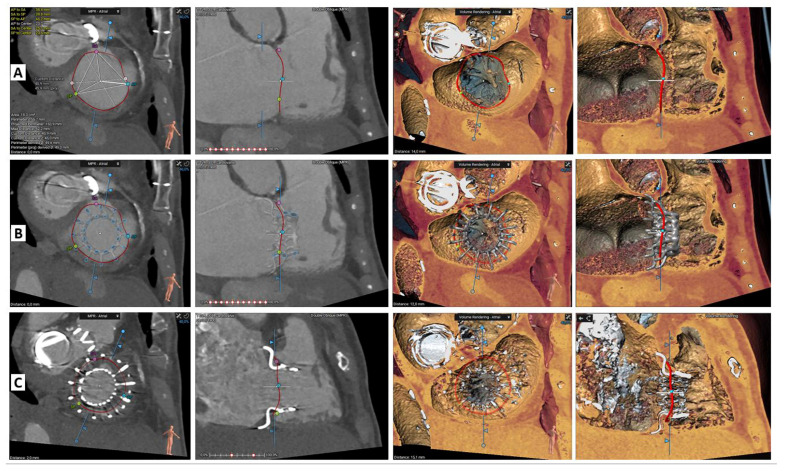
Cardiovalve TTVR case. Computed tomography pre-procedural assessment (**A**), device deployment simulation (**B**), and result after procedure (**C**). Tricuspid annular plane on 2D (first column) and 3D volume rendering (third column). Long-axis right ventricle on 2D (second column) and 3D volume rendering (fourth column).

**Figure 4 jcm-12-01371-f004:**
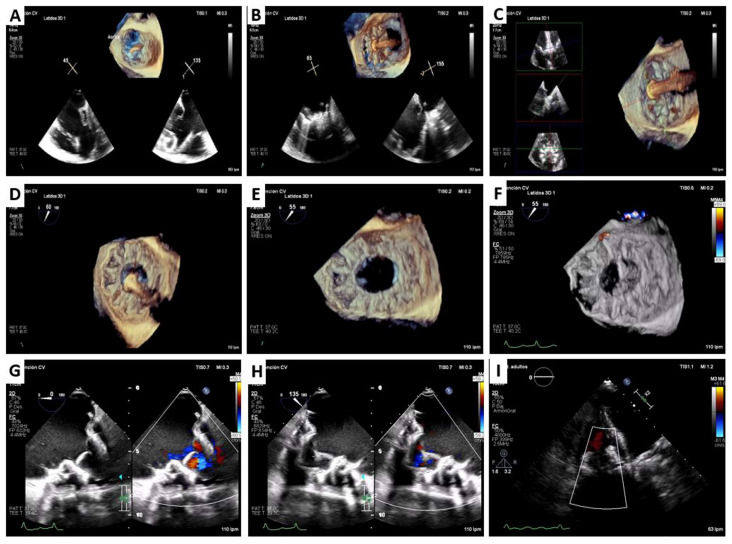
Cardiovalve TTVR case. Transesophageal echocardiography procedural monitoring. A pigtail catheter was used to advance a high support guidewire to right ventricular apex (**A**). Once the system was above the valve, the leaflet grasping legs were opened in the left atrium (**B**). The whole system was centered and advanced into the right ventricle in order to grasp the native leaflets, which was assured with live 3D-MPR (**C**). Then, atrial flange was released (**D**) and the delivery system was retrieved (**E**,**F**). The absence of tricuspid regurgitation may be checked on transesophageal (**G**,**H**) and transthoracic echocardiography (**I**).

**Figure 5 jcm-12-01371-f005:**
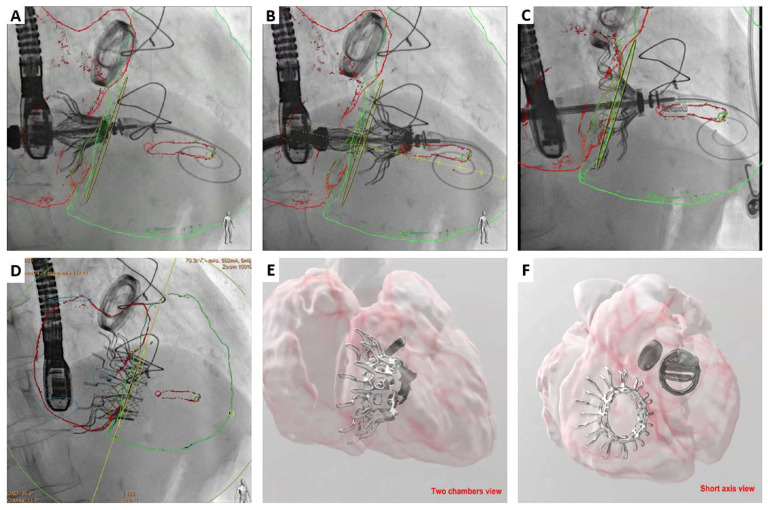
Cardiovalve TTVR case. Fluoroscopy-CT tomography fusion imaging during TTVR procedure. Leaflet grasping legs opened in the left atrium (**A**), and into the right ventricle in order to grasp tricuspid leaflets (**B**). Atrial flange released (**C**) and result after release the valve (**D**). Computed tomography 3D-segmentation 6 months after the procedure on a two-chambers view (**E**) and short axis view (**F**).

**Figure 6 jcm-12-01371-f006:**
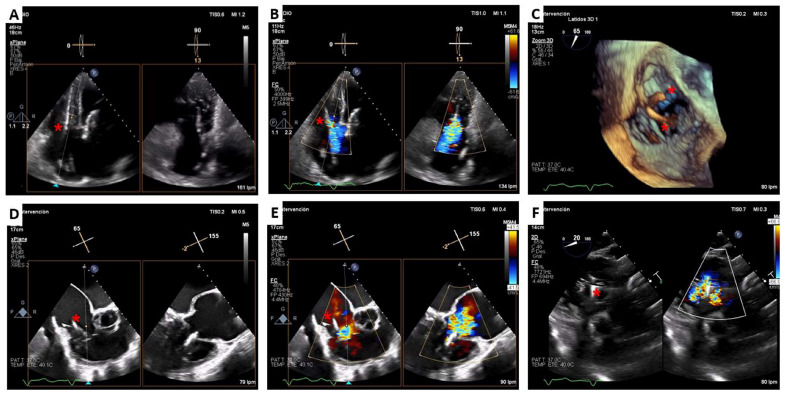
LuX-Valve Plus case. Pre-procedural transthoracic and transesophageal echocardiography. Torrential tricuspid regurgitation (**B**,**E**,**F**) secondary to annulus dilatation and leads interference (**A**,**C**,**D**). Red star: abandoned VVI pacemaker lead and functionally ICD lead.

**Figure 7 jcm-12-01371-f007:**
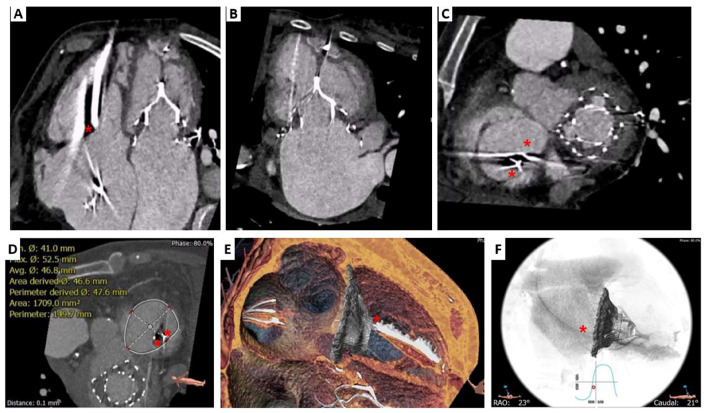
LuX-Valve Plus case. Pre-procedural computed tomography. Previous Tendyne TMVR System (Abbott Vascular, Santa Clara, CA, USA) on four chambers (**A**), two chambers (**B**) and basal short-axis (**C**) views. Tricuspid annulus assessment in mid-late diastolic phase (**D**) and virtual valve deployment (**E**,**F**). Red star: abandoned pacemaker and functionally ICD leads.

**Figure 8 jcm-12-01371-f008:**
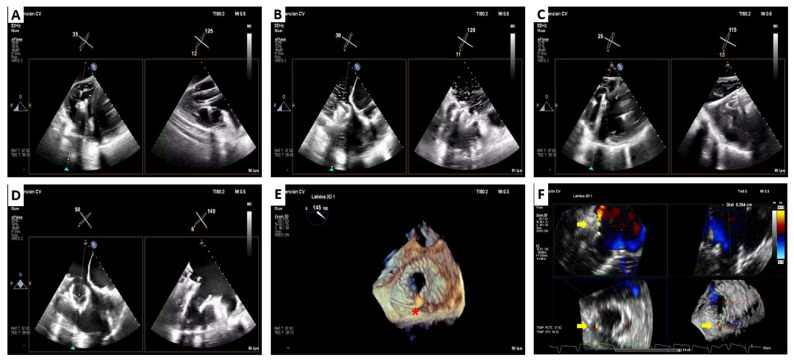
LuX-Valve Plus case. Transesophageal echocardiography procedural monitoring. Delivery system was placed perpendicular to the annular plane in a central position (**A**). The valve stent and the two anterior leaflet graspers (**B**) were withdrawn gradually. After confirming a satisfactory position, the anchoring system was released (**C**). Result: functionally and well-positioned valve (**D**), with the ICD lead placed into the posteroseptal commissure ((**E**), red star) and a mild paravalvular leak on this position ((**F**), yellow arrow).

**Figure 9 jcm-12-01371-f009:**
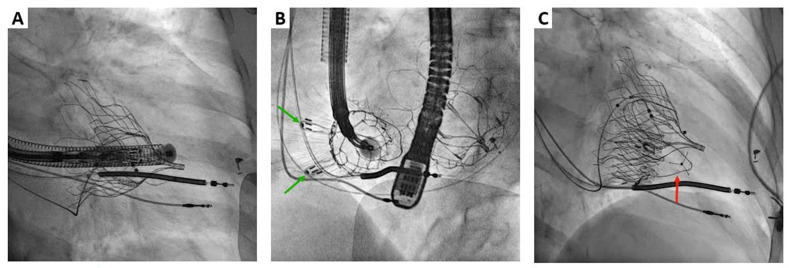
LuX-Valve Plus case. Fluoroscopic procedural monitoring. Adequate delivery system position to initiate prosthesis deployment (**A**). Orthogonal fluoroscopic projections of the deployed valve (**B**,**C**) (Green arrows: leaflet graspers. Red arrow: interventricular septal anchor).

**Table 1 jcm-12-01371-t001:** Transcatheter tricuspid valve replacement devices with larger reported experience in the clinical setting.

Device Name (Manufacturer)	Design	Access	Delivery Sheath Size	Anchoring Mechanism	Pacemaker Implant	Procedural Success in Trials	Conversion to Open Heart Surgery	Reduction of TR to Mild or Less
Gate (NaviGate Cardiac Structures, Inc.)	nitinol self-expanding conical stent with a tri-leaflet equine pericardial valve	Transjugular or transatrial	42 Fr	Native leaflets grasping	N/A	87%	5%	76%
Cardiovalve (Cardiovalve, Lda)	self-expanding nitinol frame with 3 bovine pericardial leaflets	transfemoral	28 Fr	Native leaflets grasping	N/A	N/A	N/A	N/A
Evoque TV (Edwards Lifesciences)	self-expanding nitinol frame, bovine pericardial leaflets, and an intra-annular sealing skirt	transfemoral	28 Fr	Annulus, leaflets and chords	13%	94%	1%	99%
LuX-Valve (Jenscare Biotechnology)	self-expanding stented valve, composed by a tri-leaflet bovine pericardium valve, with an external nitinol stent covered with a layer of polyethylene terephthalate	Transapical or transjugular	33 Fr	Anterior leaflet grasping and septal anchor	0%	100%	0%	83%

**Table 2 jcm-12-01371-t002:** Procedural and clinical outcomes of transcatheter tricuspid valve replacement devices with larger reported experience.

	Navigate	Evoque		LuX-Valve	LuX-Valve Plus
	Hahn et al.	Fam et al.	Kodali et al.	Mao et al.	Zhang et al.
Baseline characteristics					
Number of patients	30	25	56	15	10
Age, years (IQR)	78 (70–80)	76 (73–79)	79.3	62 (56–78)	70
NYHA III or IV, nº (%)	85.7%	88.0%	87.5%	15 (100)	10 (100)
TR severity					
Moderate	2 (6.6)	4 (16.0)	8.9%	0 (0.0)	0 (0.0)
≥Severe	28 (92.4)	25 (100)	91.1%	15 (100)	10 (100)
Procedure					
Procedural success, nº (%)	26 (87.0)	23 (92.0)	NA	15 (100)	10 (100)
Conversion to open surgery, nº (%)	2 (6.6)	0	1 (1.8)	0 (0.0)	0 (0.0)
Device embolization or malposition, nº (%)	4 (13.3)	1 (4.0)	2 (3.6)	0 (0.0)	0 (0.0)
30-day outcomes					
Mortality	3 (10)	0 (0.0)	1 (1.8)	1 (6.6)	0 (0.0)
Stroke	1 (3.3)	0 (0.0)	0 (0)	0 (0.0)	0 (0.0)
Major or life-threatening bleeding	10 (30)	3 (12.0)	1 (1.8)	1 (8)	0 (0.0)
Others	1 death for uncontrolled bleeding, 1 multiorgan failure, 1 after surgical conversion	2 pacemaker implantations	1 cardiovascular death, 2 reinterventions, 1 major access site or vascular complication	1 death related to pulmonary infection	1 pacemaker implantation for a complete AV block
TR post-procedural					
Mild or less	67.0%	88.0%	98.1%	85.7%	10 (100)
Moderate	14.0%	8.0%	1.9%	7.1%	0 (0.0)
≥Severe	19.0%	4.0%	0 (0.0)	NA	0 (0.0)
1 year outcomes		Webb et al.			
Mortality (30d-1 year)		7.4%		0 (0.0)	
Stroke				0 (0.0)	
Major or life-threatening bleeding				0 (0.0)	
NYHA I or II, nº (%)		70.0%		76.8%	
Others		2 HF hospitalizations, 1 patient new pacemaker implantation		1 device thrombosis	
TR 1 year					
Mild or less		87.0%		85.7%	
Moderate		9.0%		NA	
≥Severe		4.0%		NA	

IQR: interquartile range. TR: tricuspid regurgitation. NYHA: New York Heart Association. AV: atrio-ventricular. HF: heart failure.

## Data Availability

Not applicable.

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
