# Peer review of "Transcatheter Tricuspid Valve Replacement: Illustrative Case Reports and Review of State-of-Art"

_jcm, 2023, doi:10.3390/jcm12041371_

Round 1

Reviewer 1 Report

The manuscript, consisting of 3 case reports and review on transcatheter tricuspid valve replacement is up- to- date and clinically very important as the interest in tricuspid valve is overwhelming at the present time, moreover, transcatheter interventions on this valve sufficiently expand the group of patients whose status markedly may improve after the procedure.

The case reports are written in a clear precise manner with  9 illustrative figures.

Transcatheter tricuspid valve interventions portofolio and clinical results reviews are comprehensive and detailed, including 2 tables. However, there are no information, concerning the limitations of the described valves as well as the comparative analysis of the limitations and advantages of the described valves.

Conclusion is consistent with the data presented.

Specific comments-1). line 412-in the sentence”The device was implanted via transatrial…..” the word “approach” is lacking.

2).in the Table 2 Device embolization or malposition, nº (%) is noted in 100% of pts with LuX-Valve Plus, however this situation is not discussed or explained in the text.

The English level of the manuscript is high. The manuscript will be interesting to the wide auditorium of medical specialists, especially cardiologists, interventionalists, cardiac surgeons, etc.

Author Response

Thank you for your assessment and comments.

The authors included a short comment about comparative information of the different devices. 
"A limitation of this review is that the information available about these devices is limited, it comes from first-in-man experience or feasibility trials in many cases, and short follow-ups concerning only a few patients have been published. Nowadays, there is not enough experience to position one of the designs over the others, or to indicate its implantation in the clinical setting out of the trials or the compassionate use".

Specific comments were resolved:

  • "Approach" was added
  • % of device embolization is 0%. It was edited on the chart. 

Reviewer 2 Report

This paper presents the case of two patient underwent transcatether tricuspid regurgitation repair with novel device, specifically designed for th tricuspid valve. the paper is interesting and the following minor comments needs to be addressed.

page 1 line 35. please add more reference of TR treatment. there aother device from edwards, the cavi procedure with teh TricValve system. Please support this sentence with additional references.

page 1 line 36. in-silico modelin is emerging as a tool for the preoperative planning of transcatether treatemnts. In-silico was widely adopted to study TAVI and also TMVR, please add a sentence and the following reference.

1.Dowling C, Gooley R, McCormick L, Firoozi S, Brecker SJ. Patient-specific Computer Simulation: An Emerging Technology for Guiding the Transcatheter Treatment of Patients with Bicuspid Aortic Valve. Interv Cardiol. 2021 Aug 19;16:e26. doi: 10.15420/icr.2021.09. PMID: 34721665; PMCID: PMC8419845.

2. Pasta S, Cannata S, Gentile G, Di Giuseppe M, Cosentino F, Pasta F, Agnese V, Bellavia D, Raffa GM, Pilato M, Gandolfo C. Simulation study of transcatheter heart valve implantation in patients with stenotic bicuspid aortic valve. Med Biol Eng Comput. 2020 Apr;58(4):815-829. doi: 10.1007/s11517-020-02138-4. Epub 2020 Feb 6. PMID: 32026185.

page 8 line 15. The tricvalve system is another device to treat the TR by implanting two devices in caval position. Please add a comment on this device and similar approach where current device as the SAPIEN 3 are placed in caval position to treat TR.

Author Response

Thank you for your assessment and your comments. 

The paper is focused on transcatheter TV replacement in an orthotopic position. That's the reason to do not mention the CAVI treatment with Sapiens, TricValve, Tricento, Trillium or similar devices. We have added a sentence in the introduction to clarify this issue.

"Between these techniques, transcatheter TV replacement in an orthotopic position is an emergent and exciting field. We  present three clinical cases with two different innovative systems and a review of the state-of-art of this emergent topic."

A sentence about in-silico modelling and the references proposed were included. 
"Also, some imaging advances, such as in-silico modelling for preprocedural planning are emerging in the TV field, as there are accepted for planning before transcatheter replacement in the aortic or mitral position [4,5]."